# Control the Temperature: Selective Sampling for Diverse and High-Quality LLM Outputs

**Sergey Troshin**[*,1]**, Wafaa Mohammed**[*,1]**, Yan Meng**[*,1]**,**
**Christof Monz**[1]**, Antske Fokkens**[2]**, Vlad Niculae**[1]
[1] Language Technology Lab, University of Amsterdam
[2] Computational Linguistics and Text Mining Lab, Vrije Universiteit Amsterdam

## Abstract

Diversity is an essential metric for evaluating the creativity of outputs generated by language models. Temperature-based sampling is a common strategy to increase diversity. However, for tasks that require high precision, e.g., mathematical reasoning, uncontrolled high temperature sampling, e.g., min-$p$ or top-$p$, degrades reasoning quality. We demonstrate that the loss of accuracy is caused by sampling incorrect continuations in sensitive decoding positions. To address this, in this paper, we propose **selective sampling**, a method that dynamically switches between greedy and high-temperature sampling based on a sampling risk metric. This risk metric estimates the likelihood of output errors when applying high-temperature sampling on the current token position. To predict sampling risk, we train a lightweight classifier on a small subset of verifiable problems. The trained classifier can be integrated with the base language model with minimal latency overhead. Experiments on mathematical reasoning tasks demonstrate that selective sampling enhances the quality-diversity trade-off, even in high-temperature settings.

## 1 Introduction

Recently, large language models (LLMs) have demonstrated unprecedented capabilities in mathematical reasoning tasks by efficiently harnessing task-specific rewards (OpenAI et al., 2024; Dubey et al., 2024; Yang et al., 2024; DeepSeek-AI et al., 2025). To maintain high reasoning accuracy, prior work has primarily focused on deterministic decoding, e.g., greedy decoding, which generates outputs with the highest probability. However, this approach often leads to pathological solutions by over-optimizing for a single objective, i.e., accuracy, at the expense of other desirable properties (Hashimoto et al., 2019). In particular, deterministic decoding tends to reduce the diversity in generated outputs (Le Bronnec et al., 2024; Kirk et al., 2024). This is problematic because human preferences are not always aligned with those of greedy generations due to a lack of diversity and fluency (Zhang et al., 2021; Holtzman et al., 2020).

Diversity is important for controllability in large language models: by having access to a probabilistic sampler, a user can first generate multiple outputs, and then select the best ones using a task-specific metric (Stiennon et al., 2020); a user can improve parallel exploration of potential solutions (Yao et al., 2023), or control the generation process from a black-box sampler according to additional constraints (Mudgal et al., 2024; Deng & Raffel, 2023; Troshin et al., 2024). Temperature sampling is a common strategy to increase creativity and diversity in many LLM inference frameworks such as vLLM (Kwon et al., 2023) or Hugging Face (Wolf et al., 2020). However, it often comes at the cost of lower task accuracy compared to deterministic decoding (Shi et al., 2024). This trade-off has motivated research on how to narrow the quality gap between deterministic and temperature sampling (Holtzman et al., 2020; Basu et al., 2021; Hewitt et al., 2022; Minh et al., 2025).

---

[*]Authors contributed equally; correspondence to `serj.troshin@proton.me`

Previous sampling methods, such as top-$p$ and min-$p$ sampling, truncate the next-token distribution based on model confidence in order to improve the quality-diversity trade-off in language generation. These sampling methods prioritize high-likelihood candidates to improve output precision (Meister et al., 2023). However, one limitation of the truncation sampling approaches is that they solely rely on model confidence to sample potential candidate tokens. While the shape of the model output distribution over candidate tokens can represent uncertainty, it is hard to distinguish between different types of uncertainty (Baan et al., 2023), namely between variability due to numerous plausible continuations (Giulianelli et al., 2023) and uncertainty about what the correct answer is. For example, if a model assigns high probability to two different answers, are both of these answers plausible, or is the model not confident which one is correct in a given context? As we demonstrate in §3, sampling at certain decoding time steps can lead to incorrect continuations in cases where the greedy continuations are correct. This shows that it is important to adopt different decoding strategies across decoding positions to balance quality and diversity.

To balance both the diversity and quality of LLMs output, we propose **selective sampling**, a method that dynamically switches between greedy and temperature sampling based on a *sampling risk* metric. This metric measures the likelihood of output errors for a given decoding timestep when applying temperature sampling to it. We train a lightweight classifier on a small subset of verifiable problems to predict sampling risk. Our approach preserves the original model outputs and is easy to implement, with the classifier integrating into the base language model with minimal latency overhead. We empirically demonstrate that temperature sampling with our classifier results in a better quality-diversity trade-off compared to commonly used truncation-based and entropy-based sampling methods, both under standard and high-temperature settings.

The structure of our paper is as follows: In §2 we review the fundamental theoretical concepts behind the sampling methods explored in our study. In §3, we highlight the gap we aim to fill by analyzing when existing sampling methods fail. In §4, we detail the intuition and implementation of the selective sampling method. In §5 we outline the details of our experimental setup, including models, tasks, and evaluation metrics. Then, in §6, we present and discuss our results and findings. We describe related work in §7. Finally, we summarize our conclusions in §8.

## 2 Preliminaries

In this section, we cover key concepts for sampling mechanisms, including temperature sampling and its variants.

### 2.1 Temperature Sampling

At each step of decoding, the base model observes an already generated prefix $x$, and predicts the logits $z_{\text{LM}}(\cdot|x) \in \mathbb{R}^{|V|}$. Logits are then used to form a sampling distribution $p(v|x)$, which may involve filtering, temperature, and other modifications (Minh et al., 2025; Holtzman et al., 2020; Basu et al., 2021; Hewitt et al., 2022). In particular, *temperature sampling* uses a scalar parameter to rescale the probabilities to make the distribution more peaked (low temperature) or flatter (high temperature). The next token is then sampled from a categorical distribution defined by $p(v|x)$.

### 2.2 Temperature Sampling Variants

There are many heuristics for selecting a short-list of most likely token candidates, and they rely on different statistics of the model distribution. **Top-k** always selects top $k$ most probable tokens (Fan et al., 2018). **Top-p** (Holtzman et al., 2020) selects top tokens whose cumulative probability exceeds a hyperparameter $p$. Recently, Minh et al. (2025) proposed **min-p** that uses the discounted probability of the top-1 token to define the probability threshold. $\epsilon$**-sampling** (Hewitt et al., 2022) allows any token with a probability greater than $\epsilon$. Some of the methods rely on the Shannon entropy of the model distribution (Shannon,

**Question:** Yara qualified for a spot on the soccer team ... and a water bottle for ¥3. How much did she spend on athletic gear?
**Answer: 115**

**CoT Continuation:** Yara bought a jersey for ¥24. The leggings cost ¥13 more than the jersey… So the cleats cost __

*Probability*

**Greedy** ✅ : … So the cleats cost **20**. The headbands cost ¥11 and she got a ¥1 discount … The final answer is **115**

**Non-Greedy** ❌: … So the cleats cost **30** - 20 = 10. The headbands cost ¥11 and she got a ¥1 discount …The final answer is **105**

*Vocabulary*

20  30  1 … the  50  6

**Figure 1:** High sampling risk example for the current decoding position. **CoT Continuation** is generated by greedy decoding. **Greedy** continuation 20 results in a correct answer while **non-greedy** continuation 30 leads to an incorrect answer. The full example is shown in Appendix G, Figure 15.

1948), among which $\eta$-**sampling** truncates words below an entropy-dependent probability threshold (Hewitt et al., 2022). Zhang et al. (2024b) propose **EDT** (Entropy-based Dynamic Temperature) sampling in which they dynamically adjust the temperature based on entropy. These methods generally lead to better results than sampling directly from the unmodified $p(v|x)$, and improve diversity and accuracy. In §6, we find that even the best among them still exhibit quality degradation when increasing the temperature. We provide a more detailed description of the aforementioned methods in Appendix C.

## 3 When Do the Sampling Methods Fail?

In this section, we analyze the limitations of widely used sampling methods introduced in §2 by focusing on their impact on specific decoding time steps. We aim to identify sensitive positions where sampling significantly increases the risk of generating incorrect outputs compared to greedy decoding. To quantify this, we propose a metric *sampling risk* that measures the likelihood of output errors when replacing greedy tokens with sampled ones. Our findings highlight the need for adaptive decoding strategies that switch selectively between greedy and sampling to maintain output quality.

### 3.1 Sampling Risk

Motivated by the concept of *regret* in reinforcement learning (Bubeck & Cesa-Bianchi, 2012), we want to know the risk of choosing a sampling method in the current state compared to greedy decoding. We define **sampling risk** of a current prefix $x$ as follows:

$$\text{s-risk}(x) := R(x) - \mathbb{E}_{v \sim p}\left[R\left([x, v]\right)\right], \tag{1}$$

where $[x, v]$ denotes the concatenation of the current prefix with a sampled next token, and $R(x)$ is the reward obtained by continuing from $x$ with only greedy tokens until the stopping criteria is met. In our case, we use accuracy as a reward.[1] A higher sampling risk value indicates a greater likelihood of generating incorrect outputs when applying temperature sampling for this decoding time step.

### 3.2 Case Studies of High Sampling Risk

To validate the definition of the sampling risk, we first conduct a case study to investigate whether we can identify certain decoding time steps with high sampling risk. For this study, we focus on an arithmetic reasoning task. We hypothesize that when a model produces an arithmetic calculation, there are certain decoding steps with high sampling risk.

---

[1]Accuracy is measured by matching between predicted and gold final answers.

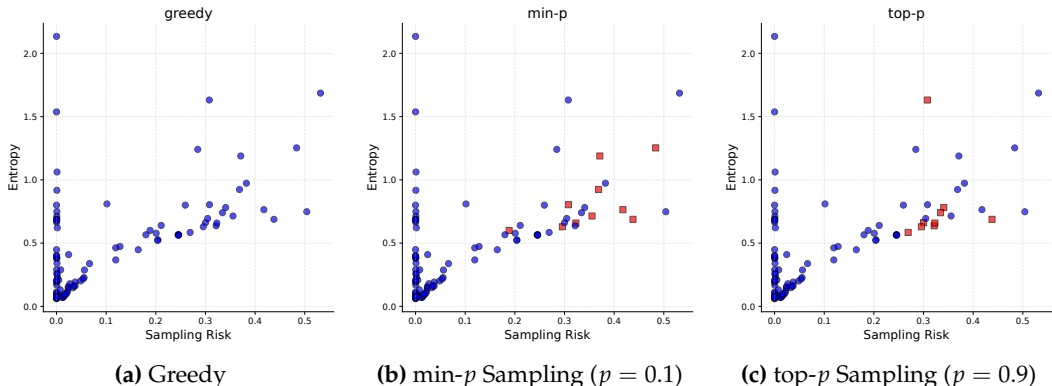

**(a)** Greedy      **(b)** min-$p$ Sampling ($p = 0.1$)      **(c)** top-$p$ Sampling ($p = 0.9$)

**Figure 2:** Entropy *vs.* Sampling Risk. Red squares: Incorrect final answers; Blue dots: Correct final answers. Overall, we show that introducing sampling methods for high-risk token positions will lead to erroneous outputs.

### 3.2.1 Setup

We focus on an arithmetic reasoning task with the chain-of-thought GSM-Symbolic dataset (Mirzadeh et al., 2025), and use the instruction-tuned LLaMa-3.1-8B model (Grattafiori et al., 2024) as our language model. For this analysis, we subsampled 100 correct greedy outputs from the CoT GSM-Symbolic dataset to ensure our observations were not limited to a few cases. Within these outputs, we identified potentially risky token positions considering the positions where the model's top-1 token is an integer number. These positions are critical for math tasks, as integer numbers often serve as intermediate results for arriving at the final answer.

### 3.2.2 Findings

Figure 1 illustrates one high sampling risk example from the GSM-Symbolic dataset. It shows that selecting a non-greedy token (e.g., 30) for the current greedy CoT continuation leads to an incorrect answer. This suggests that sampling under the model distribution at the selected integer positions tends to generate errors, as only a narrow set of candidates can yield correct outputs.

To further examine the impact of sampling at the selected positions, we compare min-$p$ and top-$p$ sampling with greedy decoding. Figure 2 shows the correlation between sampling risk and entropy for different decoding settings. We focus on full CoT continuations that produce correct answers with greedy decoding and compute sampling risk and entropy at integer positions within these continuations. Overall, we show that there are certain decoding time steps when sampling tends to result in incorrect outputs, which also correlate with high model entropy. These results underscore the importance of the sampling risk metric in guiding decoding methods to improve the quality-diversity trade-off.

## 4 Methodology: Selective Sampling

Motivated by the observation that high sampling risk leads to incorrect predictions (§3), in this section, we propose the **selective sampling** strategy by training a classifier $s_\theta(x) \in \mathbb{R}_+$ to discriminate between safe and high-risk prefixes $x$.

### 4.1 Estimating Sampling Risk

We automatically label sampling risks defined in Equation (1) for a small training set to train a classifier for selective sampling. For the training set, we either use a part of the training splits of the considered problems (GSM8k, Minerva Prealgebra tasks), or we split the test set into train and test (GSM-Symbolic task) in a proportion 60 : 40. For the GSM-Symbolic task,

> **Question:** Find the sum of all positive integers $n$ such that $1.2n-4.4<5.2$
> **Answer: 28**

**S-risk labels:** We have that $1.2n-4.4<5.2$, so $1.2n<5.2+4.4=9.6$, and thus $n<\frac{9.6}{1.2}=8$. Therefore, the positive integers $n$ satisfying the inequality are $1,2,3,4,5,6,7$. The sum of these integers is $1+2+3+4+5+6+7=\boxed{28}$. Final Answer: The final answer is $28$. I hope it is correct.

**Our classifier:** We have that $1.2n-4.4<5.2$, so $1.2n<5.2+4.4=9.6$, and thus $n<\frac{9.6}{1.2}=8$. Therefore, the positive integers $n$ satisfying the inequality are $1,2,3,4,5,6,7$. The sum of these integers is $1+2+3+4+5+6+7=\boxed{28}$. Final Answer: The final answer is $28$. I hope it is correct.

**Figure 3: S-risk labels**: obtained sampling risk labels for the Minerva task validation set; **Our classifier**: corresponding predictions from our classifier. Orange marks the high sampling risk positions where the greedy action would be chosen.

we split the data using the original problem IDs, such that versions of the same problem only appear in one subset.

For a given task, we assume that a training example contains a source prompt $u$, and we can estimate sampling risk for a given decoding time step. We do not use any correct chain-of-thoughts from the data, and only use the model outputs. We only use correct greedy outputs for training. We mark a prefix $x$ to be either risky or safe to sample from as follows:

$$y(x) = \begin{cases} 1, & \text{if s-risk}(x) < 1 - \epsilon, \\ 0, & \text{otherwise} \end{cases}, \qquad (2)$$

where we set $\epsilon = 0.05$ in our experiments.

In practice, we observe that for the reasoning tasks we considered, greedy sampling often produces high-quality outputs. Given a set of training prompts $U = \{u\}_{i=1}^N$, we obtain the subset of prompts and *correct* greedy continuations $U_+ = \{(u, x) | u \in U, R(u) = 1\}$. Then, for every $(u, x) \in U_+$, we estimate sampling risks up to a maximal token position $M$ (we use at most $M = 300$). To estimate the sampling risk at position $i$ for the current $x_{:i}$, we create a short-list of the top $k$ most probable next tokens. We force decode each of the $k$ next token candidates $\{v_j\}_{j=1}^k$, and then we finish each candidate using greedy decoding to estimate $R([x, v_j])$. Here, we treat greedy continuation as a low-cost approximation to the upperbound on the quality given a selected next token. Different ways to estimate this property may be possible and may depend on the task or the domain.

## 4.2 Selective Sampling

Our goal is to train a classifier to predict the sampling risk based on a representation of the model's context. LLMs' hidden representations are known to be a rich source of information for feature extraction (Duan et al., 2024; Mahaut et al., 2024). To obtain high-quality context features for the classifier, we simply use the hidden representations of the last token position. Namely, for a prefix $x = [x_1, \cdots, x_t]$, we use hidden states of the base model from the residual stream, including the last input embedding, for the last position $t$: $f(x) = [h_1, \cdots, h_L]$, where L is the number of layers in the base model. Then, the classifier is applied on top of the frozen hidden states $f(x)$. We parametrize a simple linear binary classifier $s_\theta(f(x)) = \sigma(\frac{1}{L} \sum_{i=1}^L w_i^T h_i)$, where $\sigma(y) = 1/(1 + e^{-y})$. Our classifier is a simple linear model applied on top of the hidden states, introducing minimal computational overhead. Most inference frameworks support access to the last hidden states, and we implement it directly inside the vLLM implementation of the LLaMa model (Kwon et al., 2023). During inference, we use our selective sampling classifier together with a truncation sampling (we use min-p, $p = 0.1$). The aim is to investigate how our classifier can complement the limitations of truncation sampling (§3).

We train our classifier in a teacher-forcing regime by passing the inputs to the model together with target sampling risks. We use binary cross-entropy loss to train the classifier. We discuss the training details in the Appendix A. Our implementation is released as open-source.[2]

---

[2]https://github.com/serjtroshin/selective_sampling

## 5 Experiments

We conduct a comprehensive evaluation of selective sampling, comparing it to existing sampling methods across multiple tasks. Our experiments aim to show the effectiveness of selective sampling in improving both the quality and diversity of the model's outputs, particularly at higher temperatures.

### 5.1 Tasks

Following Grattafiori et al. (2024); OpenAI et al. (2024); Chen et al. (2024b), we evaluate on commonly used mathematical reasoning benchmarks:

- **GSM8K** (Cobbe et al., 2021): A grade school math problem-solving benchmark.
- **GSM-*Symbolic*** (Mirzadeh et al., 2025): An extended variant of GSM8K with symbolic templates, designed to provide a more reliable assessment of reasoning ability.
- **Minerva MATH** (Hendrycks et al., 2021): A dataset of competition-level mathematical problems. We conduct experiments on the PreAlgebra subset.

We follow the standard prompting formats implemented in Evaluation Harness (Gao et al., 2024). For GSM8k and GSM Symbolic, we use the standard 8-shot chain-of-thought prompting configuration. For Minerva MATH, we use the 4-shot chain-of-thought prompting configuration. Additionally, we experiment on MMLU-Pro question answering task (Wang et al., 2024) with multiple choice answers and CoT (Appendix F.1).

### 5.2 Model

We use the instruction-tuned LLaMa-3.1-Instruct (8B) model (Grattafiori et al., 2024), selected for its strong performance across various reasoning tasks.

### 5.3 Evaluation Metrics

**Quality-diversity trade-off evaluation.** We evaluate the model's output based on two key aspects: **quality** and **diversity**, using 25 samples per prompt. Quality is measured by the accuracy of the final answer, indicating whether the model generates correct solutions. For diversity, we compute the averaged *distinct n-grams* (Li et al., 2016), which quantifies the proportion of unique $n$-grams relative to the total number of $n$-grams in the generated responses, details in Appendix E. We follow Minh et al. (2025) and measure diversity only over correctly generated samples in order to be less biased towards low-quality outputs.

To present results, we vary the temperature parameter and plot the quality metric on the $x$-axis and the diversity metric on the $y$-axis. The best method would have both high-quality and diverse predictions. To compare different methods, we look at the gap between the diversity-quality plots, e.g., which method is better in terms of quality given a certain diversity value and vice versa. In order to provide an aggregated metric integrating out the temperature parameter, we also report the area under the quality-diversity plot (**AUC**) using the trapezoidal rule[3] implemented in scikit-learn (Pedregosa et al., 2011).

**Fluency evaluation.** To demonstrate the effectiveness of our method in generating diverse yet coherent samples, we evaluate the noisiness of samples generated at high temperatures. We use perplexity as a proxy for noise, with higher perplexity indicating a higher likelihood of incoherent or nonsensical text, as suggested by previous studies (Zhang et al., 2024a; Ankner et al., 2025; Marion et al., 2023). Perplexity scores were computed using the Llama-2-7B-chat-hf model.[4] For each task, we analyze a random subset of 100 instances, generating 25 samples per instance and calculating the average perplexity of the samples as the score

---

[3] We take the theoretical $(0, 0)$ point to belong to all curves even if it is not realized by any hyperparameters in the experiment.

[4] https://huggingface.co/meta-llama/Llama-2-7b-chat-hf

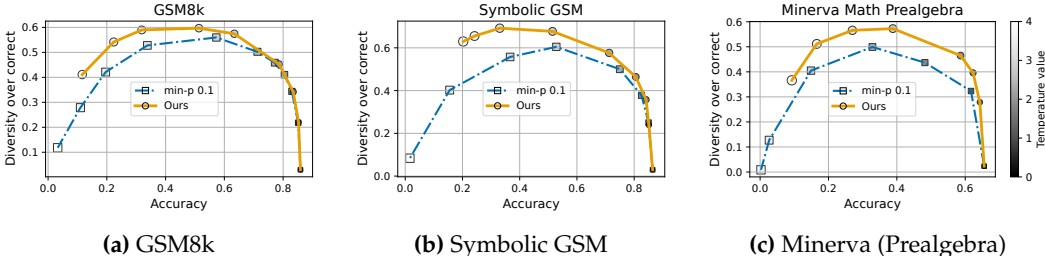

**Figure 4:** Selective sampling with our classifier improves the diversity-quality trade-off compared to the strong min$-p$ truncation baseline. On the x-axis, we report the accuracy, and on x-axis, we report the Diversity over correct samples. Size and color of the circles mark the temperature parameter.

for that instance. The overall perplexity score is then computed as the average of the per-instance scores.

## 6 Results

### 6.1 Diversity-Quality Results

We compare our selective sampling with baseline methods, including min-p sampling (Minh et al., 2025), top-p sampling, top-k (Fan et al., 2018), (Holtzman et al., 2020), $\epsilon$ sampling (Hewitt et al., 2022), $\eta$ sampling (Hewitt et al., 2022) and EDT Zhang et al. (2024b). The full plots are listed in Figures 13 and 14 (Appendix G), where we demonstrate that the baselines methods usually follow the same diversity-quality trajectory, and that there is no clear winner among the baseline methods. In Appendix D, we analyze the effect of the hyperparameters for the baselines, and we choose min-$p$, with $p = 0.1$, as the main baseline for comparison. In Table 1, we compare the cumulative aggregated diversity-quality scores (AUC metric) for the quality-diversity plots.

In Figure 4, we demonstrate that the quality-diversity trade-off of min-p can be improved using selective sampling for all three tasks. When the temperature is lower ($\tau \leq 0.5$), the performance gap between min-p and ours is small. However, our method performs better than min-p when the temperature value is increased.

Moreover, we estimate the average percentage of token positions where selective sampling chooses greedy over temperature sampling, as shown in Table 2. We observe that selective sampling tends to choose greedy decoding more often under higher temperature values and harder tasks, such as Minerva. This observation is expected, as higher temperatures or more challenging tasks tend to degrade the quality. Using greedy decoding more frequently in such cases results in a more reliable strategy to maintain the output quality.

### 6.2 Fluency Results

Figure 5 illustrates the average perplexity scores across different temperature settings for GSM8K, Symbolic GSM and Minerva Prealgebra. It also highlights the average accuracy of the samples at each temperature value. As shown in the figure, our method consistently produces significantly less noisy and more accurate samples, even at high temperatures, compared to the min-$p$ sampling method. Examples of samples generated by our method versus min-$p$ at $\tau = 2.0$ are presented in Appendix G.

### 6.3 Ablation Studies

**Classifier features.** Above, we demonstrated that our classifier trained on top of the last hidden states can improve the quality-diversity trade-off. While our classifier uses more information from the context, compared to the baselines, it is built on top of the model's

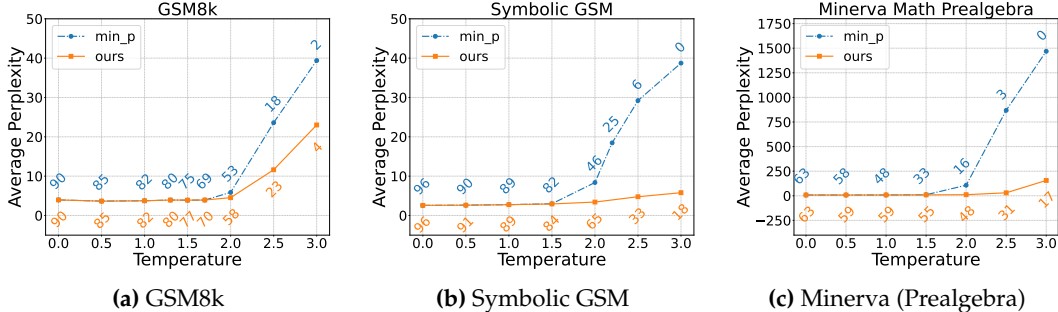

**Figure 5:** Our approach produces low perplexity samples and maintains better quality than the min-$p$ baseline even at high temperatures. Numbers inside the plots represent accuracies (as percentages).

| Method | GSM8k | Symbolic GSM | Minerva Prealgebra |
|---|---|---|---|
| top-$p$ | 0.32 | 0.32 | 0.21 |
| min-$p$ | 0.38 | 0.40 | 0.25 |
| top-$k$ | 0.38 | 0.40 | 0.23 |
| $\eta$ sampl. | 0.37 | 0.40 | 0.24 |
| $\epsilon$ sampl. | 0.37 | 0.40 | 0.24 |
| EDT | 0.35 | 0.36 | 0.24 |
| Ours | **0.42** | **0.47** | **0.30** |

**Table 1:** The area under the quality-diversity plot of various sampling strategies: an aggregated metric of the quality-diversity trade-off. Our method outperforms the baselines on this metric.

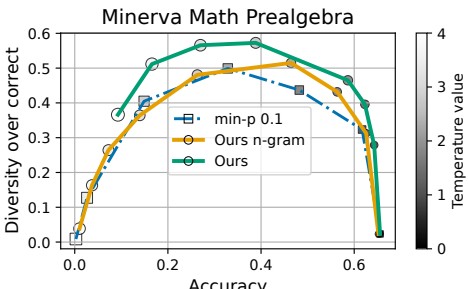

**Figure 6:** We show that our n-gram model, which does not use any internal model representations, can also slightly improve upon the min-p baseline.

| Temperature | GSM8k | Symbolic | Minerva |
|---|---|---|---|
| 1.0 | 0.02 | 0.10 | 0.37 |
| 2.0 | 0.05 | 0.12 | 0.39 |
| 3.0 | 0.17 | 0.20 | 0.44 |

**Table 2:** Percentage of greedy decoding token positions under different temperatures and tasks for selective sampling. Under higher temperatures or harder tasks, selective sampling chooses greedy decoding more than high-temperature sampling.

hidden representations. To complement the main experiment and to check whether we can train the classifier solely using the current context as the feature, we introduce a simple n-gram-based model, a classifier trained over the last $n$ input embedding representations. We provide the architectural details in Appendix B. For the Minerva evaluation set, the n-gram classifier obtains 79% accuracy and 0.72% ROC AUC, which is worse compared to the classifier trained on the hidden states (85% accuracy, and 0.78 ROC AUC). In Figure 6, we present the results for selective sampling with the n-gram classifier, where we demonstrate that the n-gram classifier is slightly better compared to the min-$p$ baseline, in term of diversity-quality trade-off, for the smaller temperature values range $\tau \in (0; 2]$. However, the n-gram classifier does not reach the performance of our hidden states classifier. We think it is easier for the model trained on top of the hidden states to generalize to the unseen contexts, especially in the higher temperature regime.

**Classifier sensitivity.** To assess the sample complexity of training the classifier, in Appendix F.2 we compare performance with random subsamples of the training set. We observe that performance convergens at around 500 training prompts, suggesting that variations in training data do not pose problems to the framework.

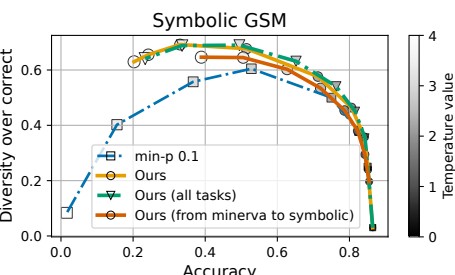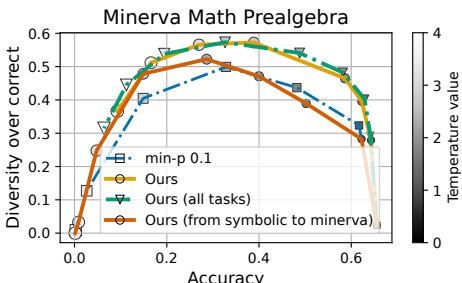

**Figure 7:** Generalization experiment with evaluation on Symbolic GSM and Minerva Math (Prealgebra) tasks. "Ours (all tasks)" is a classifier trained on all 3 datasets. "Ours (from X to Y)" denotes a transfer experiment, where we train the classifier on task X and test on Y.

**Diversity metric choice.**    To verify that our results are not overdependent on the choice of the diversity metric, we follow the RFT (Yuan et al., 2023) and use the diversity metric as the average normalized Levenshtein distance between all pairs of correct responses and compare our approach to min-p sampling. We find that our method improves the quality-diversity trade-off, in line with the findings from the n-gram-based diversity evaluation (see Figure 11 in Appendix F.3).

### 6.4    Entropy-based (EDT) baseline.

In this experiment, we modify the EDT (Zhang et al., 2024b) approach, where we use a binary threshold similar to our approach to switch between greedy and high-temperature settings. From the additional results (see Figure 12 in Appendix F.4), we observe that the threshold-based entropy baseline does not outperform the entropy-based dynamic temperature sampling (EDT), and our method outperforms both variants of entropy-based sampling approaches, which highlights the benefit of the trained classifier head versus using entropy.

### 6.5    Task Generalization of the Selective Sampling Classifier

In this experiment, we ask (1) whether we can transfer a classifier trained on one task to another task, (2) whether we can train a single classifier to be applied on many tasks. Regarding (1), we observe that selective sampling trained on the Minerva dataset outperforms the min-p baseline for the GSM Symbolic task on diversity-quality, suggesting that our classifier can generalize between these tasks (see Figure 7). Regarding (2), we train a single classifier, Ours (all tasks), on 800 examples from each of the 3 datasets: GSM8k, GSM Symbolic, and Minerva, and evaluate the model on the GSM Symbolic task. We observe that the quality of the Ours (all tasks) model closely matches that of Ours trained only on GSM Symbolic (see Figure 7, left). The same effect is observed when we evaluate the same all tasks model on the Minerva dataset (see Figure 7, right). This shows that we can use the same single classifier on multiple tasks.

## 7    Related work

In this section, we review prior work relevant to our study, focusing on sampling techniques from large language models (LLMs), methods for adjusting sampling parameters, and approaches to mitigating hallucinations. These areas collectively informed our approach for improving the quality and diversity of LLM outputs.

**Sampling from LLMs.**    In the context of sampling from LLMs, various techniques (Holtzman et al., 2020; Basu et al., 2021; Hewitt et al., 2022; Minh et al., 2025) are proposed to improve the fluency, coherence, reduction of repetitiveness, and diversity of LLM responses.

Renze (2024) finds that in context of problem solving through LLMs there is little difference between the existing sampling methods for $[0, 1]$ temperature range. Wang & Zhou (2024) observe that LLMs can reason through sampling by considering alternative decoding paths. Minimum Bayes Risk (MBR) decoding (Kumar & Byrne, 2004) improves the quality of the model by countering the pathological behavior of MAP inference (Freitag et al., 2022; Suzgun et al., 2023; Wu et al., 2025; Jinnai et al., 2024).

**Adjusting LLM sampling.** There are several relevant approaches to improve sampling quality and creativity by adjusting the temperature of the model. Liu et al. (2024) propose to dynamically choose sampling hyperparameters based on the context, where they use a prompted LLM to predict the hyperparameters at each step in a zero-shot regime. They assume LLMs already understand sampling risk. Their approach requires maintaining two models during the inference, which doubles the computation. Chang et al. (2023) develop a dynamic temperature sampling to improve the context grounding for question answering tasks, and Zhang et al. (2024b) improve upon this work by dynamically adjusting the temperature parameter based on the base model entropy. In our work, we use the approach of Zhang et al. (2024b) as an entropy-based baseline. In the context of code generation, Zhu et al. (2024) notice that challenging code tokens tend to appear at the beginning of a code block and propose to adjust the temperature based on the model confidence.

**Mitigating LLM hallucinations.** A broader related line of work improves the quality of LLMs output by mitigating hallucination (Guerreiro et al., 2023; Duan et al., 2024; Mahaut et al., 2024; Chen et al., 2023). To detect hallucinations, Duan et al. (2024) and Mahaut et al. (2024) use LLM hidden states to estimate its reliability. Chen et al. (2024a) and Meng et al. (2025) adjust the next token prediction distribution to improve the factuality and overall quality by using the model entropy. Kossen et al. (2024) propose semantic entropy probes to approximate semantic entropy from the hidden states to mitigate hallucination. Moreover, Li et al. (2023b) introduce extra models to guide the decoding while Li et al. (2023a) shift the activation during the inference to improve the truthfulness of LLMs.

## 8 Conclusion

Our work investigates temperature sampling from LLMs in the context of reasoning tasks. We highlight that fully relying on model confidence might lead to low-quality outputs. To complement the confidence estimate of a base model, we introduce the *sampling risk* metric to estimate the risk of choosing a sampling action instead of greedy, based on the expected future task-specific reward. We then propose a selective sampling approach that switches between sampling and greedy decoding at each decoding time step based on the predicted sampling risk. To achieve this, we train a simple classifier to estimate the risk during inference. We demonstrate that our selective sampling strategy can improve the quality-diversity trade-off compared to the commonly used baselines, such as min-$p$ truncation sampling. We hope our work will encourage further improvements on the important quality-diversity tradeoff in language modeling.

## 9 Limitations

Our work exhibits a few limitations. First, while our classifier for selective sampling is easy to implement, the classifier is model-dependent, which makes our approach not directly transferable between different models compared to the baseline methods. Second, we focus on the mathematical reasoning tasks, because they provide a simple way to estimate correctness using verifiable rewards (accuracy of the final answers). We do not incorporate the correctness of the CoTs as part of the quality metric, which would be much harder to evaluate. We are looking forward to exploring more open-ended and creative writing tasks in the future, with more subtle variations in sampling risks. Third, for the diversity evaluation, we only use a commonly used and intuitive n-gram-based diversity measure.

## Reproducibility Statement

Our code is based on open-source software libraries for training and evaluation of LLMs: vLLM (Kwon et al., 2023), Evaluation Harness (Gao et al., 2024), Hugging Face (Wolf et al., 2020). We conduct the experiments on commonly used datasets available via Hugging Face (Wolf et al., 2020). We use the prompt formats defined in Evaluation Harness (Gao et al., 2024). Regarding the used datasets, the GSM8k dataset (Cobbe et al., 2021) is openly available under the MIT license. Minerva MATH Prealgebra subset is part of the MATH dataset (Hendrycks et al., 2021), and by using it, we refer to the Fair Use case discussed in Hendrycks et al. (2021). Due to a recent controversy about the copyright status of the MATH dataset, we can only release data or models trained on GSM8K and GSM-symbolic. We will revisit the release of the Minerva MATH part of the data if this situation changes. This may affect the reproducibility of part of our experiments (mainly those in which we compare our classifier train on top of the hidden states with its n-gram ablation). We call for more alternative evaluation datasets designed with the intention of being used for research and released with open licenses.

## Ethics Statement

While we mitigate the degradation in quality for high temperature sampling, we have not measured whether this mitigation also affects toxicity or other undesirable properties of text. We therefore recommend the same care that users should take with any other LLM generation systems. Our method depends on the definition of task reward, which does not account for all important characteristics users might value. Since sampling risk is estimated with a classifier, it could possibly be that by suppressing certain outputs our method affects the fairness and broader diversity of the outputs.

## Acknowledgments

This publication is part of the project VI.Veni.212.228 of the research program 'Veni', which is financed by the Dutch Research Council (NWO); and is part of 'Hybrid Intelligence: augmenting human intellect' (https://hybrid-intelligence-centre.nl) with project number 024.004.022 of the research program 'Gravitation' which is (partly) financed by the Dutch Research Council (NWO). Moreover, it is funded in part by the Netherlands Organization for Scientific Research (NWO) under the project number VI.C.192.080. It is also a part of the UTTER project, supported by the European Union's Horizon Europe research and innovation programme via grant agreement 101070631.

We thank Wilker Aziz, members of LTL and CLTL labs, and the reviewers for fruitful discussions and feedback.

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

# Appendix

## A  Classifier Training

We train our classifier using Hugging Face.[5] We train our classifier with the Adam optimizer (Kingma & Ba, 2015) with $\beta_1 = 0.9, \beta_2 = 0.95, \epsilon = 1e{-}12$. We use learning rate 0.001, weight decay 0.01, and batch size 100, number of epochs 50. We use binary cross-entropy loss. We freeze all parameters of the base model and only train classifier weights. To create the labelled dataset, we use only $k = 8$ next token candidates to limit the complexity of the dataset creation, which is equivalent to using top-k filtered $p$ distribution for the regret estimation, and we use the temperature $\tau = 3.0$) to estimate the sampling risk.

For preliminary experiments, we measure the quality of the binary classification with our hidden states classifier using the validation set (100 samples from the training set). For GSM8k/Minerva our classifier obtains 93%/85% accuracy, and 0.73/0.78 ROC AUC. We visualize the predictions of our model along with original labels in Figure 3. Overall, we observe that our classifier approximates well the labels in the dataset. We hypothesize that further data refinement and noise reduction can benefit classifier training. We report dataset statistics in the Table 3 (for training, we leave only the samples with a correct greedy solution).

| Subset | GSM8k | GSM Symbolic | Minerva |
|--------|-------|--------------|---------|
| train  | 4487  | 2300         | 893     |
| val    | 100   | 100          | 100     |
| test   | 1319  | 2000         | 871     |

**Table 3:** Dataset sizes for GSM8k, GSM Symbolic, and Minerva

## B  N-gram Classifier

We implement the *n*-gram classifier as a feed-forward network applied on top of the last *n* input embeddings (we chose $n = 10$ based on the classification quality on the validation set). The network consists of the $1D$ Convolution layer with $m = 256$ filters, which aggregates the last *n* embeddings followed by the LayerNorm (Ba et al., 2016), and the GeLU nonlinearity (Hendrycks & Gimpel, 2023). Then we apply an $m \times m$ linear layer following by the the LayerNorm and the GeLU, and finally followed by the last linear layer to project into single number.

## C  Sampling Methods

### C.1  Truncation Sampling Methods

Temperature samping modifies the logits by introducing the temperature parameter $\tau$: $z_{\text{LM}}^\tau(\cdot|x) = \tau \cdot z_{\text{LM}}(\cdot|x)$. Truncated temperature sampling as implemented in commonly used frameworks (vLLM; Kwon et al., 2023) works by considering only short-list of token

---

[5]https://huggingface.co/docs/transformers/en/main_classes/trainer

candidates $V' \subseteq V$ at each decoding step (Minh et al., 2025; Holtzman et al., 2020; Basu et al., 2021; Hewitt et al., 2022) :

$$z(v|x) = \begin{cases} z_{LM}^{\tau}(v|x), & \text{if } v \in V', \\ -\infty, & \text{otherwise.} \end{cases} \quad (3)$$

and the next token is sampled from the categorical distribution:

$$\tilde{p}^{\tau}(v|x) = \frac{\exp(z(v|x))}{\sum\limits_{v' \in V'} \exp(z(v'|x))}. \quad (4)$$

**Min-$p$ sampling.** To maintain coherence at high temperatures, Minh et al. (2025) introduced **min-$p$** sampling, a dynamic method that adapts its truncation threshold according to the model's confidence at each decoding step. At each step, min-$p$ identifies the maximum probability token in the distribution:

$$p_{max} = \max_{v \in V} p(v|x). \quad (5)$$

The truncation threshold is subsequently determined by scaling a base parameter $p \in (0,1]$ by $p_{max}$: $p_{scaled} = p \times p_{max}$, and $V'$ (eq. (3)) is determined by $V' = \{v \in V : p(v|x) \geq p_{scaled}\}$

**Top-$p$ sampling.** Also known as nucleus sampling, this method samples from the "nucleus" of high-probability tokens by restricting the sampling pool to the top tokens whose cumulative probability exceeds a hyperparameter $p$ (Holtzman et al., 2020): $V' = \{v \in V : \sum p(v|x) \geq p\}$

**Top-$k$ sampling.** The sampling pool in this method consists of the most probable top $k$ tokens (Fan et al., 2018): $V' = \{v \in V : \text{rank}(p(v|x)) \leq k\}$

**$\epsilon$-sampling.** This method allows any token with a probability greater than a threshold $\epsilon$ (Hewitt et al., 2022): $V' = \{v \in V : p(v|x) > \epsilon\}$

### C.2 Entropy Sampling Methods

For language models, Shannon entropy, defined as $H[p] = -\sum_{v \in V} p(v) \log(p(v))$ measures the uncertainty in predicting the next token in a sequence (Shannon, 1948). Truncation sampling methods impose fixed thresholds to limit the set of candidate tokens during generation. While these methods can improve output quality by filtering out low-probability tokens, they may also over-restrict choices in low entropy distributions, potentially reducing diversity. Entropy sampling methods adjust the sampling process based on the model's entropy, aiming to improve the quality-diversity trade-off.

**$\eta$-sampling.** $\eta$-sampling truncates words below an entropy-dependent probability threshold (Hewitt et al., 2022): $\eta = \min(\epsilon, \alpha \exp(-H[p]))$, using $\alpha \in [0,1]$ and the hyperparameter $\epsilon$. Token shortlist is then determined as $V' = \{v \in V : p(v|x) > \eta\}$.

**EDT sampling.** Entropy-based Dynamic Temperature sampling method dynamically adjusts the temperature parameter according to the model's entropy and can be combined with truncation sampling methods as an initial step (Zhang et al., 2024b). With the recommended hyperparameters $\alpha = 0.8$ and $\theta \in [0,1]$, they shrink the original temperature $\tau$ by $f(H[p])$, namely $T(\tau, H[p]) = \tau \alpha^{\frac{\theta}{H[p]}}$. Intuitively, when the entropy is large, the shrinkage factor $f(H[p])$ approaches 1, and when the entropy is small, it approaches 0.

## D Sampling Hyperparameter Settings

For comparing sampling methods: top-$k$, top-$p$, min-$p$, $\eta/\epsilon$, and entropy-based dynamic sampling (EDT), we report their optimal results with conducted tests on different hyperpa-

rameters in Table 4. Results are selected based on the AUC scores over the quality-diversity plot, reflecting the overall quality-diversity trade-off.

For top-$p$ sampling, we report top-$p = 0.7$ and conducted tests on $p = 0.7, 0.8, 0.9$. For min-$p$ sampling, we report min-$p = 0.1$ for all experiments and conducted tests on $p = 0.1, 0.2, 0.3$. For $\eta$ and $\epsilon$ sampling, we test $\epsilon$ and $\eta$ values 0.0002 and 0.0009, found 0.0009 to score better for both values and report this in our experiment results.

For EDT sampling, we report $\theta = 0.1$ for all experiments, and conducted tests on $\theta = 0.1, 1.0$.

## E    Averaged Distinct N-gram Diversity

Given a set of test instances $S = \{s\}_{i=1}^{M}$, each with a response set $R_K = \{r\}_{i=1}^{K}$. Following Minh et al. (2025), we first filter out incorrect samples for each instance in order to be less biased towards low-quality outputs, and we end up with a set of $C$ correct samples $R_C = \{r\}_{i=1}^{C}$ per instance, where $C$ may vary across instances. Then, for $n \in \{1, 2, 3, 4, 5\}$, we calculate the averaged distinct n-gram diversity for each instance as follows:

$$\text{averaged distinct n-gram} \ (s) = \sum_{n=1}^{5} \frac{\text{set(n-gram}(R_C))}{\text{n-gram}(R_C)}.$$

The total test-set diversity is the average diversity over all instances:

$$\text{diversity}(R_C) = \sum_{i=1}^{M} \text{averaged distinct n-gram} \ (s_i).$$

## F    Additional Experiments

### F.1    MMLU (Social Tasks)

We perform an additional experiment on the MMLU-Pro question answering task (Wang et al., 2024) with multiple choice answers and CoT. We choose the subset of following tasks: law, philosophy, history, psychology, which are more different from the math tasks from our main experiments. As we can see from the results in Figure 8, and Figure 9, (a) our method slightly outperforms the min-p baseline in terms of diversity-quality trade-off and (b) better preserves the quality for higher temperatures, which broadens the potential scope for our method.

### F.2    Sensitivity to the training dataset

To verify that our classifier training is not too sensitive for the variations in training data, we perform an additional experiment for classifier training by randomly subsampling 100, 500, 1000 examples of the training set for the GSM Symbolic task. We measure the accuracy for sampling risk classification on the validation set. From Figure 10, we observe that subsampling 1000 or 500 examples reduces accuracy marginally from 0.9 to 0.89. Taking 100 examples leads to a slight overfitting and 0.87 accuracy.

### F.3    Ablation of the diversity metric

Following RFT (Yuan et al., 2023), we use a diversity metric as the average normalized Levenshtein distance between all pairs of correct responses and compare our approach to min-p sampling. We find that our method improves the quality-diversity trade-off, in line with the findings from the n-gram-based diversity evaluation (see Figure 11).

### F.4    EDT ablation experiment

In this section, we perform an additional ablation experiment for the modified the EDT (Zhang et al., 2024b) approach, where we use a binary threshold similar to our approach to

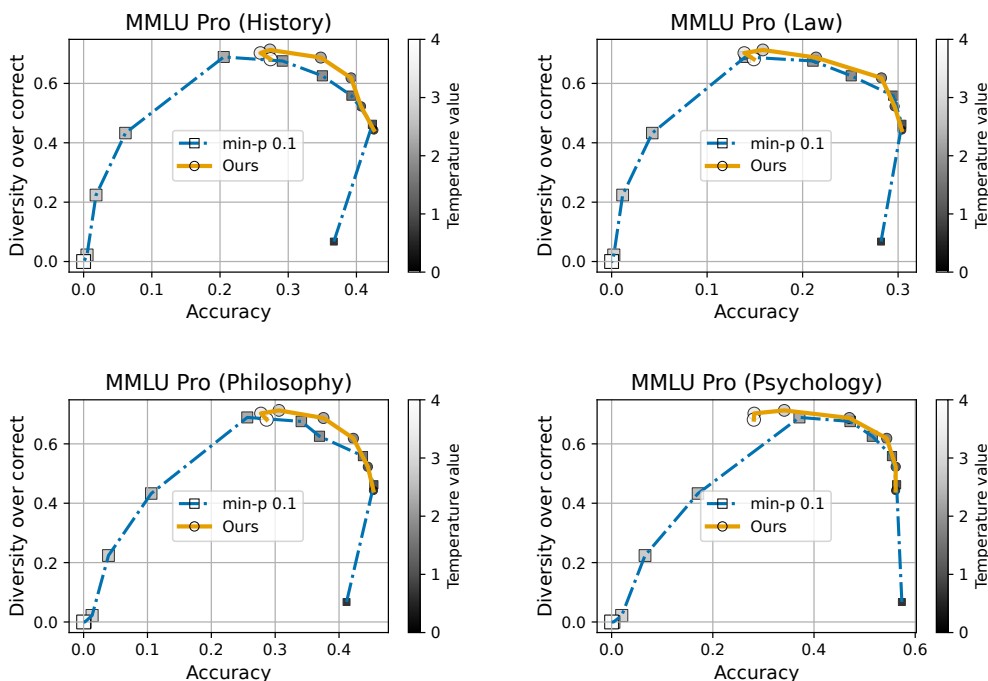

**Figure 8:** Additional result on MMLU Pro task on 4 social subsets.

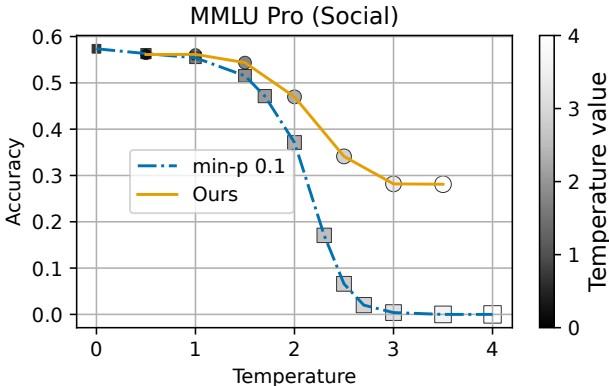

**Figure 9:** Additional result on MMLU Pro task on 4 social subsets: quality-temperature trade-off.

switch between greedy and high-temperature settings. In particular, given a threshold $t$, we evaluate the threshold-based entropy baseline (if $H[p] < t$, then $\tau'=0.0$, else $\tau'=\tau$), using $t$. from [0.5, 1, 2]. From the additional results Figure 12, we observe that the threshold-based entropy baseline does not outperform the entropy-based dynamic temperature sampling (EDT). Our method outperforms both variants of entropy-based sampling approaches, which highlights the benefit of the trained classifier head versus the entropy.

# G Examples

In Figure 15, we provide a high sampling risk example from the Symbolic GSM task. In Figure 16, we provide additional examples with $\tau = 2.0$ for the samples for GSM8k. Overall, we observe that for higher temperatures, our methods produces more fluent outputs.

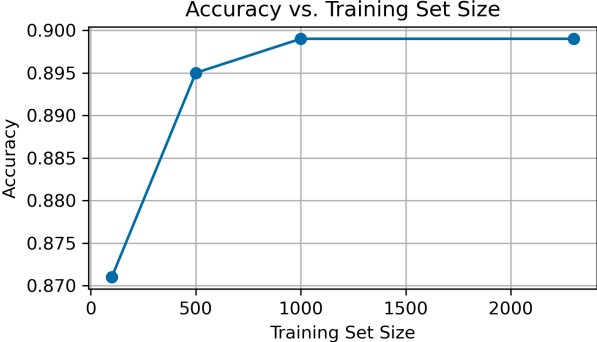

**Figure 10:** Training data sensitivity analysis on the GSM Symbolic task: accuracy vs the size of the training set.

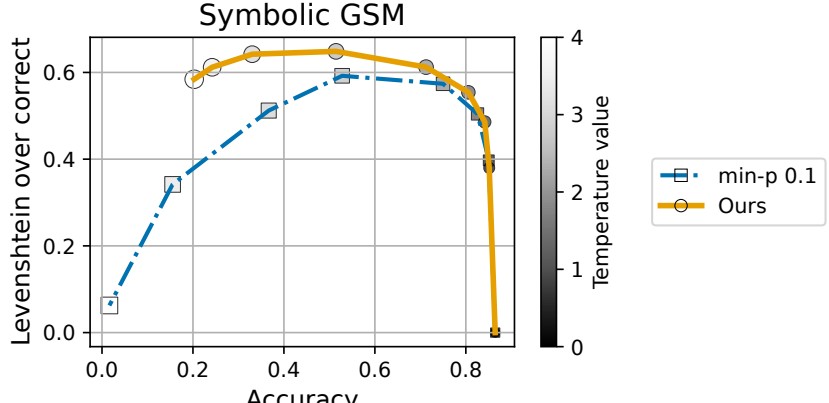

**Figure 11:** Quality-diversity trade-off plot with Levenshtein diversity metric.

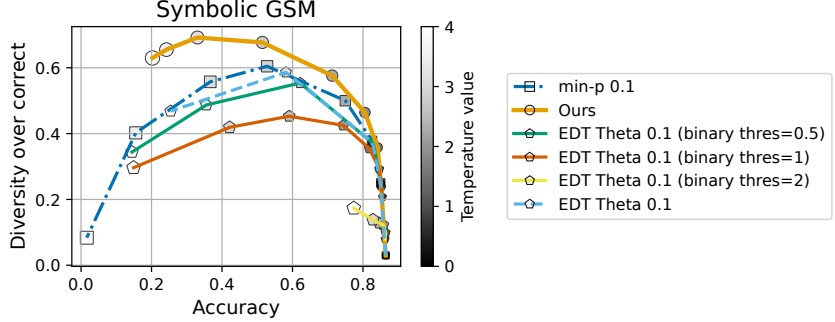

**Figure 12:** Quality-diversity trade-off plot with Levenshtein diversity metric.

| Method | Diversity-Quality **AUC** | |
| --- | --- | --- |
| | **GSM8K** | **Minerva** |
| top-k 20 | 0.38 | 0.23 |
| top-p 0.7 | 0.28 | 0.20 |
| top-p 0.8 | 0.31 | 0.20 |
| **top-p 0.9** | 0.32 | 0.21 |
| **min-p 0.1** | 0.38 | 0.25 |
| min-p 0.2 | 0.37 | 0.25 |
| min-p 0.3 | 0.37 | 0.25 |
| $\epsilon$-sampl. 0.0002 | 0.35 | 0.23 |
| **$\epsilon$-sampl.** 0.0009 | 0.37 | 0.24 |
| $\eta$-sampl. 0.0002 | 0.36 | 0.23 |
| **$\eta$-sampl.** 0.0009 | 0.37 | 0.24 |
| **EDT** $\theta = 0.1$ | 0.35 | 0.24 |
| EDT $\theta = 1.0$ | 0.32 | 0.24 |

**Table 4:** Baselines hyperparameter values. Diversity-quality AUC Scores for GSM8K and Minerva Prealgebra. Bold denotes the chosen hyperparameter value for main results.

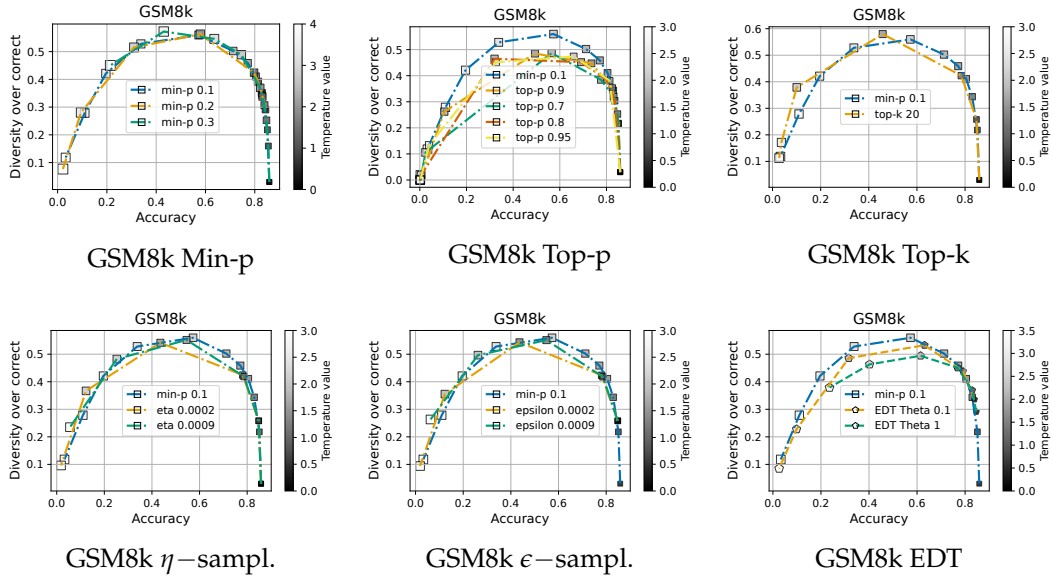

**Figure 13:** We compare different hyperparameters for the baseline methods on the GSM8k task.

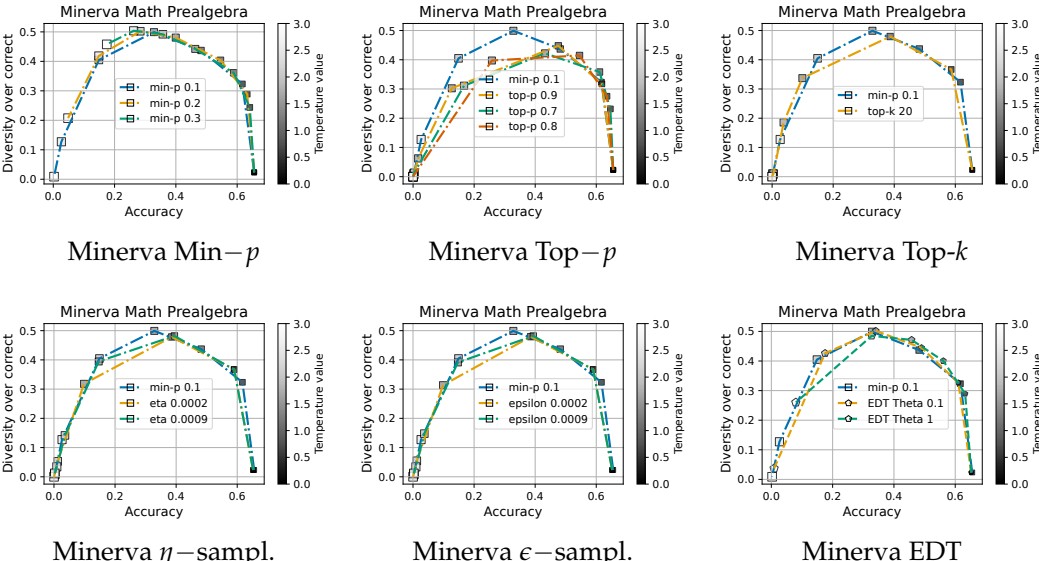

**Figure 14:** We compare different hyperparameters for the baseline methods on the Minerva task.

```
Question: Yara qualified for a spot on the soccer team, so she went shopping for some athletic gear. She bought a jersey
for ¥24, a pair of soccer leggings for ¥13 more than the jersey cost, and a pair of cleats that were originally ¥30 but
were on sale for two-thirds price. She had a coupon for ¥1 off the package of ¥11 athletic headbands that she also bought.
Finally, she got a duffle bag for ¥42 but with a 50% discount, and a water bottle for ¥3. How much did she spend on
athletic gear?
Target: 115
```

**CoT Risk Token Position: 71**

```
Yara bought a jersey for ¥24. The leggings cost ¥13 more than the jersey, so they cost 24 + 13 = 37. The cleats were
originally ¥30 but were on sale for two-thirds price. Two-thirds of 30 is (2/3) * 30 = 20. So the cleats cost __
```

**P (20 | Prefix) = 0.68**

```
Yara bought a jersey for ¥24 … Two-thirds of 30 is (2/3) *
30 = 20. So the cleats cost 20
```

**P (30 | Prefix) = 0.32**

```
Yara bought a jersey for ¥24 … Two-thirds of 30 is (2/3) *
30 = 20. So the cleats cost 30
```

**Correct Answer**

```
Yara bought a jersey for ¥24 … Two-thirds of 30 is (2/3) *
30 = 20. So the cleats cost 20. The headbands cost ¥11 and
she got a ¥1 discount, so they cost 11 - 1 = 10. The duffle
bag was originally ¥42 but she got a 50% discount, so it
cost 42 * (1/2) = 21. The water bottle cost ¥3.
Now we add up the costs: 24 + 37 + 20 + 10 + 21 + 3 = 115.
The final answer is 115
```

**Incorrect Answer**

```
Yara bought a jersey for ¥24 … Two-thirds of 30 is (2/3) *
30 = 20. So the cleats cost 30 - 20 = 10. The headbands cost
¥11 and she got a ¥1 discount, so they cost 11 - 1 = 10. The
duffle bag was originally ¥42 but she got a 50% discount, so
it cost 42 - (42/2) = 42 - 21 = 21. The water bottle cost
¥3. So the total cost is 24 + 37 + 10 + 10 + 21 + 3 = 105.
The final answer is 105
```

**(a)** Example 1

```
Q: Chen operates the cash register exactly seven times as fast as her less-experienced colleague Uma. Roya, being more
experienced than Uma but less experienced than Chen, operates at the same speed as the average of Uma and Chen. Daily, Uma
processes 22 customers. What is the total weekly production for the three if they work all days of the week?
A: 1848
```

**Risk Token Position: 103**

```
Since Uma processes 22 customers per day, and there are 7 days in a week, Uma's weekly production is 22 * 7 = 154
customers.
Chen operates 7 times as fast as Uma, so Chen's daily production is 22 * 7 = 154 customers. Chen's weekly production is
154 * 7 = 1078 customers.
Roya operates at the same speed as the average of Uma and Chen. The average of Uma and Chen's daily production is ( __
```

**P (22 | Prefix) = 0.88**

```
Since Uma processes 22 … The average of Uma and Chen's
daily production is (22
```

**P (154 | Prefix) = 0.12**

```
Since Uma processes 22 …The average of Uma and Chen's daily
production is (154
```

**Correct Answer**

```
Since Uma processes 22 … The average of Uma and Chen's
daily production is (22 * 7 = 154 customers. Chen's weekly
production is 154 * 7 = 1078 customers.
Roya operates at the same speed as the average of Uma and
Chen. The average of Uma and Chen's daily production is (22
+ 154) / 2 = 88 customers. So Roya's daily production is 88
customers. Roya's weekly production is 88 * 7 = 616
customers.
The total weekly production for the three is 154 (Uma) +
1078 (Chen) + 616 (Roya) = 1848 customers. The final answer
is 1848
```

**Incorrect Answer**

```
Since Uma processes 22 … The average of Uma and Chen's daily
production is (154 + 154) / 2 = 308 / 2 = 154 customers. So
Roya's daily production is also 154 customers. Roya's weekly
production is 154 * 7 = 1078 customers.
The total weekly production for the three is Uma's
production + Chen's production + Roya's production. 154 +
1078 + 1078 = 2310. The final answer is 2310
```

**(b)** Example 2

**Figure 15:** High-Risk sampling examples from GSM-Symbolic. **Left**: Top-1/Greedy generation on high-risk decoding position. **Right**: Non-greedy generation on high-risk decoding position.

Q. Kylar went to the store to buy glasses for his new apartment. One glass costs $5, but every second glass costs only 60% of the price. Kylar wants to buy 16 glasses. How much does he need to pay for them?
A. 64

**Ours**

**min-p**

First, let's find out which glasses cost 60%. There are 16 glasses in total and every 2nd glass counts.
 So it will be a case of the sequence, one, 60%, one, 60%. Starting with just the first one, we have a total of 8 of the 100% glasses. Since (8 * 5)= $40 For every (2nd) one there's a 60% discount so a 100% price minus (60 * 5) that we will apply 8 of those so ($5 - (60 * $5) = $1 For every 2nd glass in a pair set that is 8 times $1 so we get that as $8
Add those together - $40 + $8 we get a final of $48 For 8 of those set total that means for this set of 8 glasses are that $48 There is a total of 16 glasses
8 of those are the 100% price and the other is $48 for a total set of 8 and since its in the same ratio. 8 of $5 will be ($5) * ($8) = $40 and then $48 added together = $88 Those are (8) of (100%) and also of  $5 and the other 8 as previously calculated.
So overall in those pairs they equal ((8 * 5) + (8*1)) = $88 and (8 * 5) =  $40 So a total of 4 sets ((8 * 5)+ (8* 1)) = ((8 * 5) + (8 * 1) ) * (4) to get total glasses price which equals (($88)) * 4 ($88)) * (4)  ($88 *4) for total and that we get: $352

Kylar wants to buy 16 glasses. For every second glass, which are the even number glasses, the price is 60% or 0.60 * $5 = $3. For every odd number glass the price stays the same, at 5.
We first count the total glasses for which we pay the full price ($5). Those are 8 (from the original set) odd numbers from 1 to 16: 1,3,5,7,9,11,13,15.
8 even numbers cost the half ($3), so 8 x $3 is $24, or
8 odd numbers also costs the full ($5), 8 x 5 is 40. Now all of $24  & $40 should be combined.
 Combining those sums, you have  $64

One glass is $5, and the second glass is 60% of the price, which is $5 x 0.60 = $3. To find the number of glasses in the sequence, if the first glass is #1, the second glass is #2. We can count forward the number of glasses until 16 glass and identify how many of them at which price: #1($5) - #2($3), #3($5) - #4($3), and so forth. From these, we find, that $5 is priced for all odd numbers, while $3 is priced for the even numbers in the counting of the set {#1-$5; #3- $5, and so forth ending with  #15-$5 } and only the even numbers {#2-$3; #4- $3, and so forth ending with #14- $3} in set {#1-$5; #2- $3} and so forth {#3- $5 ; #4- $3 }and ending #15,  and #16 ($5 and $3 respectively) glasses at $5 and $3 prices and Kylar plans to buy 8 glasses of each.
 Number of glasses:
First glasses ($5): {#1, 3, 5, 7, 9, 11, 13, 15}. Count: 8 glass.
 ($5) + (8 *$5) Total value of $5 priced glasses $45
  The second set of glasses ($3): {#2, 4, 6, 8, 10, 12, 14, 16}.
($3 * 8) for total 8  priced glasses of  value of 24 $.
 Adding value $45 (odd priced), of glass and $24 (even price of a) glass to the  total make the sum
total =$45 + $24 =$69
The final answer is $69.

First we need to find the cost for 8 normal and 8 discounted glasses separately.
 Normal glass costs 5 dollars each.
 Normal cost: 8 * 5 = 40
 Discounted glasses will each cost only 0.6 times the price. i.e each of these will be 3.
 Discounted glass cost: 8 * 3 = 24
Adding normal and discounted glasses:
 total glass cost = Normal cost + discounted glass cost = 40 + 24 = 64
The final answer is 64

First, let's find out which glasses cost 60%. There are 16 glasses in total and every 2nd glass counts.
 So it will be a case of the sequence, one on, one on for every 2.
Glass number is the position of glass when in a counting list. 2 is on place 2. On (position #), two Off
On-Position (counting#) off position # number and list begin again so can find second cost with counter function. List begin 1 as #one glasses bought

 we have position count ( Counter= glass list counter) = # on Counter = List numb in a position each cost depends when = you order like so buy first one . $  = Number= five start case…

… absurd placement poetry contributions memorial conflicting never dr communicating winner organisation begins diagram exert upgrading proven combinations practical carrying fossil popular replied echoed divergence affecting expansion screened highlight morning points clothing members silver roots forward revenge ob endurance funeral exemption exceeding retired invaded driving our cancer deserving I sought developments

Kylar wants to buy 16 glasses. For every second glass, which are the even number glasses, the price is 60% or 0.60 * $5 = $3. For every odd number glass the price stays the same, at 5.
We first count the total glasses for which we pay the full price ($5). Those are 8 (from the original set) because from 1 through 8 is half of 16 which includes the number of glass price equaling five dollars, odd and even glass together then Kylar purchases an additional 6 even glasses for $3, making the following two rows using variables
 Number and Price or Cost = (2x dollars:full price - odd)
           x additional 3 ( $ per glass the cost for additional /half glasses.
Using just Kylar needs purchase/already known even odd ones listed order into solution!)
        Number:2
        of Glasses using Numbers from Kylars store price ordered correct order both were full ($ 5)
             {even Cost 6 remaining remaining another }….

… resulting etc dollars higher bought going four are go can answer start divided then less dollar trying they learn needs want after still all half seem full can didn addition certainly term now both addition dont try best information new there needs subtract using numbers let found gave fairly over will price has will more possible such makes add adding big gave some come quite addition find problem proper above still 51 will way clearly lowest second division based * pay   all greater known came we took again so example) get clear great present usually true meaning following show ending lot numbers extra both common simpler x shown general mostly correctly prior two addition amounts under our compare below right short related buy future move we enough end information good in earlier individual divided fully today four

**Figure 16:** Example from GSM8k for Ours with temperature 2.0 versus min-p with temperature 2.0. Our model is more stable with higher temperatures compared to uncontrolled min-p, which fails co produce coherent outputs for some prompts with temperature 2. We skip part of the output for min-p since it is too long.

