# OpenReview forum: "Control the Temperature: Selective Sampling for Diverse and High-Quality LLM Outputs"
_colmweb.org/COLM/2025/Conference — COLM 2025_

### Official Review · Reviewer_YhHJ · 2025-04-22

**Rating:** 8
**Confidence:** 4
**Ethics Flag:** 1

**Summary:**

The paper proposes a decoding strategy in which the temperature is low for parts of the output that need to be exact, and high for parts where creativity in the output is appreciated.

**Questions To Authors:**

* How does the work relate to constraint decoding? Couldn't the selective sampling also be modeled as a setup with high temperature but additional constraints? It would be nice to understand the relation between these perspectives.

**Reasons To Accept:**

* clearly motivated paper
* very simple and straight-forward idea
* good experimental setup

**Reasons To Reject:**

* the experiments are only conducted on one data set, which is a math solving resource. I am wondering what the impact of the proposed strategy would be on purely creative tasks.
* it is not really clear to me why, in such setups of math solving, a high temperature should generally be preferable. Perhaps the motivation could be improved in this direction.

---

> ### Author Response · Authors · 2025-06-02
> **Authors response**
>
> Thank you for your review. We are glad to hear that our paper (1) **is clearly motivated**; and that we introduce (2) **simple and straight-forward idea**; (3) **good experiment setup**.
>
> We address your comments below and will incorporate your feedback in the final version. We link the additional results in the pdf via an anonymous link as allowed via COLM guidelines.
>
> > the experiments are only conducted on one data set, which is a math solving resource. I am wondering what the impact of the proposed strategy would be on purely creative tasks.
>
> We perform an **additional experiment on MMLU-Pro** question answering task with multiple choice answers and CoT. We choose the subset of following tasks: law, philosophy, history, psychology, which are more different from the math tasks from main experiments. As we can see from the results in Figure R1a and Figure R1b, (a) **our method slightly outperforms the min-p baseline in terms of diversity-quality trade-off** and (b) **better preserves the quality for higher temperatures**, which broadens the potential scope for our method.
> We don't test our method on creative writing, where there is no verifyable reward, since we condider such tasks as out of scope.
>
> Additional results (see Figure R1a and Figure R1b): https://anonymous.4open.science/r/rebuttle_selective_sampling-153C/Rebuttal.pdf
>
> > it is not really clear to me why, in such setups of math solving, a high temperature should generally be preferable. Perhaps the motivation could be improved in this direction.
>
> We think that enabling higher diversity of math CoT can potentially benefit parallel exploration of different solution paths, e.g. the "parallel thinking" or Tree of Thoughts approaches. As reviewer vo4Q mention, we tackle the well-recognized trade-off between generation quality and diversity in LLMs, which is a critical area of research. We promise to improve the motivation part in the introduction.
>
> > How does the work relate to constraint decoding? Couldn't the selective sampling also be modeled as a setup with high temperature but additional constraints? It would be nice to understand the relation between these perspectives.
>
> We think our selective sampling can be further combined with the commonly used controllable generation methods such as DExperts - Liu et al. 2021 or RAD - Deng et al. 2023, since they both operate on the logits of the base model (see for example Dekoninck et al 2023, Controlled Text Generation via Language Model Arithmetic for how to combine multiple constraints). If you have any particular constraint decoding direction in mind, we would be happy to discuss it further.

---

> > ### Comment · Reviewer_YhHJ · 2025-06-06
> >
> > Thank you for the answer. The additional results are really interesting. I suggest to add them to the paper, or if space does not allow, to the appendix. Great work!

---

### Official Review · Reviewer_vo4Q · 2025-05-12

**Rating:** 5
**Confidence:** 4
**Ethics Flag:** 1

**Summary:**

This paper addresses the challenge of maintaining high-quality outputs while promoting diversity in LLM generations, particularly in tasks requiring precision like mathematical reasoning. Authors show that prior temperature-based methods, while increasing diversity, often degrade reasoning quality by sampling incorrect continuations in sensitive, high-entropy positions. The authors propose a selective sampling method that dynamically switches between greedy decoding and high-temperature sampling, which is guided by a sampling risk metric, estimating the likelihood of errors if high-temperature sampling is applied at a given token position. A classifier is trained on a samll subset of verifiable problems to predict this sampling risk. Experiments on mathematical reasoning demonstrate that selective sampling improves the quality-diversity trade-off, especially in high-temperature settings, and produces less noisy samples compared to existing methods.

**Questions To Authors:**

Could you elaborate on the characteristics and size of the "small subset of verifiable problems" used for training the sampling risk classifier? How sensitive is its performance to variations in this training data?

Could you provide more context for Figure 2? Specifically, how were the examples and token positions selected for this analysis, and does it represent a few illustrative cases or a broader trend?

For the fluency experiments (Figure 5), what was the accuracy of your method compared to min-p sampling at the different temperature points shown, to better understand the quality-fluency trade-off?

The experiments use 25 samples per prompt. Given that prior research (e.g. Smaller, Weaker, Yet Better -- Bansal et al., 2024, Large language monkeys -- Brown et al., 2024) suggests diversity can continue to increase with a higher number of samples, what was the rationale for this specific number, and have you explored the impact of more samples?

Could you provide an analysis or statistic on how frequently the selective sampling model chooses greedy decoding versus high-temperature sampling across the evaluated datasets?

The classifier uses hidden states from all layers of the base model. Have alternative feature sets (e.g., a subset of layers, attention-based features) been explored, and how did they compare in performance and efficiency?

have you considered comparing selective sampling against a simpler heuristic that directly uses an entropy threshold (perhaps learned or set empirically) to switch between greedy and temperature sampling, to further isolate the benefit of the trained classifier head?

In Figure 4, the color map for temperature values and the colors used for the different methods are very similar. It would be better to use more distinct color scheme for enhanced clarity.

For GSM8K, it is insightful to analyze the diversity of the generated solutions not just by n-grams in the full CoT, but also by looking at the diversity of the underlying mathematical steps or equations (RFT -- Yuan et al., 2023). Could you explore this as well?

Do you have some insights on the "sampling risk" metric and classifier training for tasks lacking easily verifiable rewards, such as open-ended creative writing?

**Reasons To Accept:**

The paper is generally well-written, and the authors are upfront about several limitations of their work.

The paper tackles the well-recognized trade-off between generation quality and diversity in LLMs, which is a critical area of research.

The concept of "selective sampling" based on a learned "sampling risk" metric is an innovative and intuitively appealing solution.

Within the domain of mathematical reasoning, the experiments demonstrate improvement in the quality-diversity trade-off and fluency compared to strong baselines like min-p sampling.

**Reasons To Reject:**

The experiments are focused solely on mathematical reasoning tasks. While suitable for evaluating precision, the paper's claims would be significantly strengthened by demonstrating efficacy on a broader range of tasks.

The method's current framework appears to require training a separate classifier for each specific task to estimate risk. This limit generalization and usability. A dedicated generalization study showing the classifier's capabilities across different tasks, or a direction towards task-agnostic risk predictor would be very important for broader impact.

Why are different types of uncertainty important for sampling? The motivation for distinguishing between different types of uncertainty (citing Baan et al., 2023 ) and its direct impact on the necessity of selective sampling versus general uncertainty awareness could be more explicitly connected and justified.

The experimental details surrounding Figure 2, which illustrates sampling risk versus entropy, are not clear and should be expanded. It's unclear how many examples or positions are represented and the precise methodology for selecting these instances for analysis.

The rationale for estimating sampling risk by sampling only a single token $v_j$ and then completing the sequence with greedy decoding needs further discussion. While this isolates the immediate impact of $v_j$, it doesn't fully represent the behavior of continuous temperature sampling throughout a trajectory.

The sampling risk classifier is trained on "a small subset of verifiable problems". This raises questions about the method's applicability to tasks where such verifiable rewards are not easily obtainable or are more subjective.

The trained classifier is model- and task-dependent, potentially limiting its direct transferability between different base LLMs and tasks without retraining.

While EDT sampling is used as an entropy-based baseline, a more direct baseline that uses entropy as a real-time proxy to switch between greedy and temperature sampling (without the dynamic temperature adjustment of EDT but rather a binary switch similar to the proposed method) can provide a better measure of the added benefit of the trained classifier head.

The paper uses the accuracy of the final answer as the quality metric. While understandable for complexity reasons, not incorporating the correctness or plausibility of the Chain-of-Thought (CoT) itself is a limitation in fully assessing reasoning quality.


Some further concerns are in questions to authors.

---

> ### Author Response · Authors · 2025-06-02
> **Authors Response (part 1)**
>
> Thank you for your review. We are encouraged to hear that our paper (1) proposes an **innovative and intuitively appealing sampling approach**; (2) **shows improvements** in the quality-diversity trade-off compared to strong methods, e.g., min-p.
>
> We address your comments below and will incorporate your feedback in the final version. We link the additional results in the pdf via an anonymous link as allowed via COLM guidelines.
>
> > paper's claims would be significantly strengthened by demonstrating efficacy on a broader range of tasks
>
> We perform an **additional experiment on MMLU-Pro** question answering task with multiple choice answers and CoT. We choose the subset of following tasks: law, philosophy, history, psychology, which are more different from the math tasks from main experiments. As we can see from the results in Figure R1a and Figure R1b, (a) **our method slightly outperforms the min-p baseline in terms of diversity-quality trade-off** and (b) **better preserves the quality for higher temperatures**, which broadens the potential scope for our method.
>
> Additional results (see Figure R1a and Figure R1b): https://anonymous.4open.science/r/rebuttle_selective_sampling-153C/Rebuttal.pdf
>
>
> > A dedicated generalization study showing the classifier's capabilities across different tasks, or a direction towards task-agnostic risk predictor would be very important for broader impact.
>
> We conduct an additional experiment following your suggestion (additional results:  *Figure R2. Generalization experiment.*). (1) We observe that selective sampling trained on the Minerva dataset, **outperforms the min-p baseline for GSM Symbolic task** on diversity-quality, suggesting that **our classifier can generalize between the tasks** (see Figure R2a).
> (2) We train a single classifier [*Ours (all tasks)*] on 800 examples from each of the 3 datasets: GSM8k, GSM Symbolic, Minerva datasets, and evaluate the model on the GSM Symbolic task. We observe that the quality of the *Ours (all tasks)* model closely matches that of *Ours* trained only on GSM Symbolic (see *Figure R2a*). Same effect is observed when we evaluate the same *all tasks* model on the Minerva dataset (see *Figure R2b*). This shows that **we can use the same single classifier on multiple tasks**.
>
> Additional results (see Figure R2): https://anonymous.4open.science/r/rebuttle_selective_sampling-153C/Rebuttal.pdf
>
> > Why are different types of uncertainty important for sampling? The motivation for distinguishing between different types of uncertainty (citing Baan et al., 2023 )
>
> We will extend the discussion about uncertainty in the introduction. In particular, given that uncertainty of the model is expressed through one representation i.e. model logits, it is hard to distinguish cases where high production variability is possible (multiple plausible next tokens), vs high model uncertainty (model is not sure what is correct answer). Our analysis demonstrates that if the model is uncertain about which answer is correct,  this leads to higher sampling risk. Selective sampling learns to distinguish these two cases of uncertainty to prefer sampling for high production variability positions.
>
> > The rationale for estimating sampling risk by sampling only a single token
> and then completing the sequence with greedy decoding needs further discussion. While this isolates the immediate impact of vj, it doesn't fully represent the behavior of continuous temperature sampling throughout a trajectory.
>
> We observe that for the reasoning tasks we considered, greedy sampling often produces high-quality outputs. Furthermore, we use only those training examples where greedy leads to correct answer. We thus treat greedy continuation as a low-cost approximation to the upperbound on the quality given a selected next token. Different ways to estimate this property may be possible, and may depend on the task or the domain. We find our definition to be a good starting point, but we will mention this possible avenue for further research in the manuscript.

---

> > ### Author Response · Authors · 2025-06-02
> > **Authors response (part 2)**
> >
> > > Could you elaborate on the characteristics and size of the "small subset of verifiable problems" used for training the sampling risk classifier? How sensitive is its performance to variations in this training data?
> >
> > As we explain in 4.1, we first split the examples into train/test sets. Then, only for the train set, we filter out the prompts with incorrect greedy continuations, and leave 100 examples for validation. Here, we present the statistics for the filtered train/val and test sets, which we will add in the paper text.
> > | sizes     | GSM8k | GSM Symb | Minerva |
> > | ---- | ----- | -------- | ------- |
> > | train| 4487  |   2300   |   893   |
> > | val  | 100   |   100    |   100   |
> > | test | 1319  |   2000   |   871   |
> >
> >
> > To verify that **our classifier training is not too sensitive for the variations in training data**, based on your suggestion, we perform an additional experiment for classifier training by randomly subsampling 100, 500, 1000 examples of the training set for GSM Symbolic. We measure the accuracy for sampling risk classification on the validation set. From Figure R3, we observe that subsampling 1000 or 500 examples reduces accuracy marginally from 0.9 to 0.89. Taking 100 examples leads to a slight overfitting and 0.87 accuracy. We will include this ablation experiment in the final version of the paper.
> >
> > Additional results (see Figure R3): https://anonymous.4open.science/r/rebuttle_selective_sampling-153C/Rebuttal.pdf
> >
> >
> > > Could you provide more context for Figure 2? Specifically, how were the examples and token positions selected for this analysis, and does it represent a few illustrative cases or a broader trend?
> >
> > For this analysis, we subsampled 100 correct greedy outputs from the CoT GSM-Symbolic dataset to ensure our observations were not limited to a few cases. Within these outputs, we identified potentially risky token positions considering the positions where the model's top-1 token is an integer number. These positions are critical for math tasks, as integer numbers often serve as intermediate results for arriving at the final answer. We will add the additional details in the paper text.
> >
> >
> > >For the fluency experiments (Figure 5), what was the accuracy of your method compared to min-p sampling at the different temperature points shown, to better understand the quality-fluency trade-off?
> >
> > Figure 5 is calculated with the 100 randomly sampled subset of the instances used in Figure 4. In the next version of the manuscript, for each point on the temperature-fluency plot, we will include the accuracy value.
> >
> >
> > > The experiments use 25 samples per prompt. Given that prior research (e.g. Smaller, Weaker, Yet Better -- Bansal et al., 2024, Large language monkeys -- Brown et al., 2024) suggests diversity can continue to increase with a higher number of samples, what was the rationale for this specific number, and have you explored the impact of more samples?
> >
> > In our work, 25 samples per prompt is a hyperparameter of the metric, commonly used in controlled generation literature (DExperts, Liu et al. 2021; Reward Augmented Decoding, Deng et al 2023). If we increase the number of samples, we expect to observe the increase in diversity metric values, but we note that using higher number of samples will increase the costs of evaluation.
> >
> >
> > > Could you provide an analysis or statistic on how frequently the selective sampling model chooses greedy decoding versus high-temperature sampling across the evaluated datasets?
> >
> > Following your suggestion, we estimate the percentage of average number of token positions, where selective sampling chooses greedy over temperature sampling below.
> > | % greedy token positions     | GSM8k | GSM Symb | Minerva |
> > | ---- | ----- | -------- | ------- |
> > | t = 1.0 |   2%    |   10%      |     37%   |
> > | t = 2.0 |   5%   |     12%    |     39%   |
> > | t = 3.0 |   17%   |     20%    |    44%   |
> >
> > From these results, we see an interesting trend that for higher temperature values, selective sampling tends to more often choose greedy option as expected, since higher temperature negatively impacts the quality. Moreover, we see that for the harder minerva task, selective sampling tends to select greedy decoding more often. We will include these results in the final version of the paper.
> >
> >
> > > In Figure 4, the color map for temperature values and the colors used for the different methods are very similar. It would be better to use more distinct color scheme for enhanced clarity.
> >
> > This is a great suggestion, we will improve the visibility of temperature values and the color scheme in the final version.

---

> > > ### Author Response · Authors · 2025-06-02
> > > **Authors response (part 3)**
> > >
> > > > For GSM8K, it is insightful to analyze the diversity of the generated solutions not just by n-grams in the full CoT, but also by looking at the diversity of the underlying mathematical steps or equations (RFT – Yuan et al., 2023). Could you explore this as well?
> > >
> > > We agree that a finegrained diversity evaluation of the underlying steps is important. Following the suggested work (RFT – Yuan et al., 2023), we use a diversity metric as the average normalized *Levenstein distance* between all pairs of correct responses and compare our approach to min-p sampling. **We find that our method improves quality-diversity trade-off, in line with the findings from the n-gram-based diversity evaluation** (additional results: *Figure R4. Ablation of the diversity metric.*). We will include this evaluation in the final version of the paper.
> > >
> > > Additional results (Figure R4): https://anonymous.4open.science/r/rebuttle_selective_sampling-153C/Rebuttal.pdf
> > >
> > >
> > > > The classifier uses hidden states from all layers of the base model. Have alternative feature sets (e.g., a subset of layers, attention-based features) been explored, and how did they compare in performance and efficiency?
> > >
> > > In our preliminary experiments on GSM Symbolic, we tried using just last layer or middle layer. We observed that the classifier with all hiddens obtains 0.9 accuracy, the one with the last layer obtains 0.89 accuracy with slightly more unstable training, while the middle layer classifier converges to 0.88 accuracy (all on the validation set).
> > >
> > > In terms of efficiency per token, our linear classifier requires $d$ multiplications per layer, where $d$ is hidden state size. Since the mainstream architecture we use has complexity of self-attention, which is proportional to $d \cdot L$  per layer, where L is context size, we consider the complexity of our classifier negligible. This is also what we observed in preliminary experiments. However, for architectures with better base model complexity, alternative feature sets may be promising.
> > >
> > >
> > > > have you considered comparing selective sampling against a simpler heuristic that directly uses an entropy threshold (perhaps learned or set empirically) to switch between greedy and temperature sampling, to further isolate the benefit of the trained classifier head?
> > >
> > > Following the suggestion, we perform an additional experiment to evaluate the threshold-based entropy baseline (if entropy < thres., then temp=0.0, else temp=max_temp), using thres. from [0.5, 1, 2]. From the additional results (Figure R5. EDT ablation experiment), we observe that the **threshold-based entropy baseline does not outperform the entropy-based dynamic temperature sampling (EDT)**. Our method outperforms both variants of entropy-based sampling approaches, which highlights the benefit of the trained classifier head.
> > >
> > > Additional results (Figure R5): https://anonymous.4open.science/r/rebuttle_selective_sampling-153C/Rebuttal.pdf
> > >
> > > > Do you have some insights on the "sampling risk" metric and classifier training for tasks lacking easily verifiable rewards, such as open-ended creative writing?
> > > > method's applicability to tasks where such verifiable rewards are not easily obtainable or are more subjective.
> > >
> > > We are looking forward to explore more open-ended creative writing tasks in future work, where we anticipate to see more subtle variations in sampling risk. As we limit the scope of our work to the tasks with verifyable rewards, we do not experiment with creative writing tasks. We will improve the text by highlighting our assumption more clearly.

---

> > > > ### Comment · Reviewer_vo4Q · 2025-06-10
> > > >
> > > > Thank you for the detailed reply and new experiments.
> > > >
> > > > Since Minerva is considered more difficult than GSM Symbolic it would make more sense that one shows the generalization from a simpler task to a more complex one.
> > > >
> > > > Regarding the uncertainty rationale, your reply describes what your method does (distinguishing uncertainty types) but not why this distinction is fundamentally important. A more principled justification for this core concept is needed.
> > > >
> > > > Regarding "greedy sampling often produces high-quality outputs", are these observations mentioned or shown somewhere?
> > > >
> > > > Regarding Figure 5. as it stands, it still does not show the accuracies for fair comparison,
> > > >
> > > > The finding that your model defaults to greedy sampling most often on Minerva seems counter-intuitive. This limits the exploration where it might be needed most. What is the reason behind this behaviour?

---

> > > > > ### Author Response · Authors · 2025-06-11
> > > > > **Authors response**
> > > > >
> > > > > Thank you for your response. We address your additional comments below and incorporate the feedback in the final version.
> > > > >
> > > > > >Since Minerva is considered more difficult than GSM Symbolic it would make more sense that one shows the generalization from a simpler task to a more complex one.
> > > > >
> > > > > It would be indeed valuable to have a generalization from the simple task to a more complex task. Following your suggestion, we do the experiment applying a model trained on the GSM symbolic to the Minerva dataset (additional results: Figure R2b), however, we don't observe a consistent improvement for the whole temperature range. We will include this experiment in the new version of the manuscript.
> > > > >
> > > > > >Regarding the uncertainty rationale, your reply describes what your method does (distinguishing uncertainty types) but not why this distinction is fundamentally important. A more principled justification for this core concept is needed.
> > > > >
> > > > > The distinction of the different uncertainty types is part of our motivation. We ask whether that we can use model representations beyond model entropy and verify this hypothesis empirically.
> > > > >
> > > > > >Regarding "greedy sampling often produces high-quality outputs", are these observations mentioned or shown somewhere?
> > > > >
> > > > > For the problems we experiment on, the task quality of the greedy outputs is high (>60% accuracy for Minerva and >80% accuracy for GSM tasks). We will improve the phrasing in the paper.
> > > > >
> > > > > >Regarding Figure 5. as it stands, it still does not show the accuracies for fair comparison,
> > > > >
> > > > > We include the temperature-quality plots (see Figure R6 in additional results) with accuracies as text labels. We observe that selective sampling maintains the quality better than min-p baseline **both for fluency and accuracy**. We will update these plots in the next version of the manuscript as well.
> > > > >
> > > > > >The finding that your model defaults to greedy sampling most often on Minerva seems counter-intuitive. This limits the exploration where it might be needed most. What is the reason behind this behaviour?
> > > > >
> > > > > Given that the accuracy of the base model is lower for Minerva compared to GSM symbolic, we think it is expected that our classifier chooses greedy option more often.
> > > > >
> > > > > Additional results (see Figure R2b and Figure R6): https://anonymous.4open.science/r/rebuttle_selective_sampling-153C/Rebuttal.pdf

---

### Official Review · Reviewer_s7Xi · 2025-05-13

**Rating:** 7
**Confidence:** 4
**Ethics Flag:** 1

**Summary:**

The paper studies a novel sampling approach that seeks to better balance the trade off between quality (accuracy) and the creativity (diversity) of LLMs’ generations. The authors first illustrated the limitations of current temperature-based sampling approaches, such as min-p and top-p, through the lens of sampling risk. Sampling risk measures the risk of error in taking a non-greedy sampling approach at a particular decoding time step. This observation motivates the proposed approach of selective sampling, where a classifier is trained to take in the final LLM hidden states to predict the sampling risk (S-risk) at a particular time step, to toggle greedy decoding when risk is high and temperature-based sampling when risk is low, based on a threshold value. Experiments were conducted on math reasoning-based datasets, including GSM8K, GSM-symbolic and Minerva MATH.

**Reasons To Accept:**

The proposed approach is well motivated, backed by analyses on existing temperature-sampling approaches.

The paper is well-written and easy to follow.

**Reasons To Reject:**

The experiments are conducted only on math reasoning tasks. It remains to be shown if this approach is effective for other reasoning tasks. From Figure 3, it seems that the parts with high S-risk are concentrated at tokens related to digits and math operations, which suggests that this approach is relevant and effective for math-based reasoning but it is less clear how well this approach would apply to non-math questions which may not involve math-related tokens as much.

---

> ### Author Response · Authors · 2025-06-02
> **Authors response**
>
> Thank you for your review. We are encouraged by your recognition that our paper (1) proposes a **novel sampling approach** to balance the quality and diversity of LLMs' generation; (2) is **well-written** and easy to follow.
>
> We address your comments below and will incorporate your feedback in the final version. We link the additional results in the pdf via an anonymous link as allowed via COLM guidelines.
>
> > The experiments are conducted only on math reasoning tasks. It remains to be shown if this approach is effective for other reasoning tasks. From Figure 3, it seems that the parts with high S-risk are concentrated at tokens related to digits and math operations, which suggests that this approach is relevant and effective for math-based reasoning but it is less clear how well this approach would apply to non-math questions which may not involve math-related tokens as much.
>
> We perform an **additional experiment on MMLU-Pro** question answering task with multiple choice answers and CoT. We choose the subset of following tasks: law, philosophy, history, psychology, which are more different from the math tasks from main experiments. As we can see from the results in Figure R1a and Figure R1b, (a) **our method slightly outperforms the min-p baseline in terms of diversity-quality trade-off** and (b) **better preserves the quality for higher temperatures**, which broadens the potential scope for our method.
>
> Additional results (see Figure R1a and Figure R1b).
> https://anonymous.4open.science/r/rebuttle_selective_sampling-153C/Rebuttal.pdf

---

> > ### Comment · Reviewer_s7Xi · 2025-06-08
> > **Thank you for the response**
> >
> > The reviewer thanks the authors for the response and have adjusted the score accordingly.

---

### Decision · Program_Chairs · 2025-07-08

**Decision:**

Accept

**Comment:**

This paper proposes a new approach that dynamically switches between greedy sampling and temperature sampling to obtain a good balance between accuracy and diversity of the model's responses. All the reviewers generally liked the paper, and the only marginally negative reviewer had no substantial concerns. The other two reviewers provide good scores. Therefore, we recommend acceptance of the paper.